# A Systematic Review of the Effect of Osteoporosis on Radiographic Outcomes, Complications, and Reoperation Rate in Cervical Deformity

**DOI:** 10.3390/jcm14176196

**Published:** 2025-09-02

**Authors:** Ishan Shah, Elizabeth A. Lechtholz-Zey, Mina Ayad, Brandon S. Gettleman, Emily Mills, Hannah Shelby, Andy Ton, William J. Karakash, Apurva Prasad, Jeffrey C. Wang, Ram K. Alluri, Raymond J. Hah

**Affiliations:** Department of Orthopaedic Surgery, Keck School of Medicine of USC, Los Angeles, CA 90033, USA; lechthol@usc.edu (E.A.L.-Z.); mina.ayad2@uhhospitals.org (M.A.); brandon.gettleman63@gmail.com (B.S.G.); emily.mills7@gmail.com (E.M.); hshelby@usc.edu (H.S.); andyton@usc.edu (A.T.); wkarakas@usc.edu (W.J.K.); apurvapr@usc.edu (A.P.); jeffrey.wang@med.usc.edu (J.C.W.); ram.alluri@med.usc.edu (R.K.A.); ray.hah@med.usc.edu (R.J.H.)

**Keywords:** osteoporosis, cervical deformity, spine surgery, outcomes

## Abstract

**Background/Objectives:** The purpose of this review was to determine the impact of osteoporosis on outcomes after surgery for cervical deformity. Cervical deformity involves abnormal curvature or misalignment of the cervical spine, often resulting in a significant loss of quality of life and requiring surgical correction. While osteoporosis has been associated with hardware failure including screw loosening and cage migration in spine surgery, its role in cervical deformity remains unclear. Existing studies report mixed findings with regard to postoperative sequelae in patients with osteoporosis undergoing surgical correction of cervical deformity. **Methods:** A systematic review using PRISMA guidelines and MeSH terms involving spine surgery for cervical deformity and osteoporosis was performed. The Medline (PubMed) database was searched from 1990 to August 2022 using the following terms: “osteoporosis” AND “cervical” AND (“outcomes” OR “revision” OR “reoperation” OR “complication”). This review focused on radiographic outcomes, as well as post-operative complications. **Results:** Eight studies were included in the final analysis. Three papers assessed risk factors for the development of post-operative distal junctional kyphosis (DJK), but only one found osteoporosis as a predictor for DJK. Although three studies found that osteoporosis was not significantly associated with the incidence of surgical complications, one highlights osteoporosis as a predictor of complications at 90 days postoperatively (*p* < 0.001) and another associates osteoporosis with overall poor outcomes (*p* = 0.021). Furthermore, one study assessing the relationship between osteoporosis and reoperation found no association. **Conclusions:** Overall, our systematic review suggests that in patients undergoing surgery for cervical deformity, osteoporosis is not predictive of the need for reoperation or the development of postoperative complications, such as DJK, dysphagia, superficial infection, and others. These findings highlight the need for further study regarding the role of osteoporosis in surgical correction of cervical deformity.

## 1. Introduction

Cervical deformity encompasses a variety of spinal pathology involving the abnormal curvature or misalignment of the cervical spine with an estimated incidence of approximately 30% [1,2]. Presenting with a range of symptomatology and often resulting in significant impairments and reduction in quality of life, many patients opt for surgical correction as definitive management [3,4]. Indications for surgery include neurological compromise, pain, and progressive deformity. Outcomes after corrective spine surgery have been well-studied in relation to many comorbidities, of which osteoporosis and its associated reduction in bone density have been hypothesized to play a role [5,6].

The importance of rigorous investigation into the impacts of osteoporosis in spine surgery are further underscored by its large clinical burden, with an estimated global prevalence between 18–19% [7,8]. Currently, the role of osteoporosis within the context of patients undergoing surgical correction of cervical deformity remains unclear, with some studies highlighting poorer outcomes and increased complications in osteoporotic patients and others demonstrating no association. Though the proposed mechanism of decreased bone mineral density has been substantiated, further exploration is warranted due to the mix of evidence present in the current literature [9,10,11,12]. Of note, recent reviews on osteoporosis in adult spinal deformity, along with cervical and lumbar degenerative pathology, have highlighted its association with increased rates of complications including cage migration, screw loosening, and revision surgery [13,14,15]. However, given the reduced load present in the cervical spine as compared to the lumbar spine and the resultant varying complication rates, the effect of osteoporosis may vary as well. Overall, the literature studying the impacts of osteoporosis in cervical deformity has yet to be studied in aggregate.

Within the context of a large global burden, further exploration on this subject is necessary to help guide standardized clinical practices regarding preoperative screening and management of osteoporosis in patients with cervical deformity. As such, this study aimed to systematically review the literature to determine the impact of osteoporosis on outcomes and complications after surgery for cervical deformity.

## 2. Materials and Methods

A systematic review of the literature to determine the impact of osteoporosis on outcomes following surgical correction of cervical deformity was carried out following the PRISMA (Preferred Reporting Items for Systematic Reviews and Meta-Analyses) guidelines. No informed consent or IRB approval were necessary. The Medline (PubMed) database was searched from 1990 through August 2022 using the following terms: “osteoporosis” AND “cervical” AND (“outcomes” OR “revision” OR “reoperation” OR “complication”). During the initial screen, studies were included if they were written in English, both prospective or retrospective, assessed patients with a diagnosis of osteoporosis, and reported medical, surgical, radiographic, and other postoperative outcomes. A diagnosis of osteoporosis was determined based on ICD-10 coding or decreased bone density on dual-energy X-ray absorptiometry (DEXA) or computed tomography (CT).

In the initial exclusion filter, we removed studies including systematic reviews, literature reviews, case reports, and studies without osteoporotic patients or relevant postoperative outcomes. Studies where the indications for surgery were related to trauma, neoplasia, or compression fractures were also excluded. Studies with limited sample sizes (*n* < 20), along with those lacking a surgical intervention, were also removed. Following this initial screen, full-text articles were eliminated if they did not relate to cervical deformity or if cement augmentation was performed.

Two independent reviewers carried out the electronic database search and screening. Title, abstract, and full text were used to screen studies and duplicates were removed. References from the full texts were examined to determine if any relevant studies were missed within the initial search. Any studies obtained via references were subject to the screening criteria mentioned prior to ensure that solely articles which reported on outcomes after surgery for cervical deformity were included. Following study selection, data collection and systematic review were carried out.

The Cochrane Consumers and Communication Review Group’s Data Extraction Template for Included studies was used to create a data collection sheet [16]. Three authors reviewed eligible articles to collect author name, publication year, study type, patient populations, indications for surgery, and the outcomes assessed. Outcomes from each study were categorized as radiographic, surgical, patient-reported, and other complications. A risk of bias assessment was conducted using the GRADE (Grading of Recommendations Assessment, Development, and Evaluation) method. A GRADE designation of very low, moderate, or high was assigned to each article [17].

## 3. Results

### Study Characteristics

Eight studies were included in the final analysis (Figure 1). Of these, seven were retrospective reviews of prospectively collected databases, and one was a retrospective database study (Table 1). The incidence of osteoporosis in these study samples ranged from 10.9 to 19.4% (Table 2).

## 4. Radiographic Outcomes

Three articles assessed the incidence of distal junctional kyphosis (DJK) following cervical deformity correction, all of which were retrospective reviews of prospectively collected databases. The incidence of DJK in these studies ranged from 23.1 to 31.8%. A 2023 study by Passias et al. investigating predictors of DJK following surgical correction of adult cervical deformity in 110 patients found that osteoporosis conferred a higher risk of DJK after surgery, with a coefficient of 0.633 used in a model to predict the risk of this complication with an AUC of 77.8% [9]. Contrarily, two studies in 2018 and 2020 by Passias et al. also attempting to determine predictors of DJK following surgical correction of cervical deformity in samples of 101 and 117 patients, respectively, were both unable to establish significance between osteoporosis and DJK [10,20].

## 5. Complications

### 5.1. Postoperative and Clinical Outcomes

There were five retrospective studies that assessed postoperative outcomes for individuals with osteoporosis who underwent surgical intervention for cervical spine deformity (Figure 2). The incidence of any complication in these studies ranged from 20.2 to 64.2%. Two studies identified a significant difference in complication rates for individuals with osteoporosis undergoing correctional cervical spine deformity surgery. In 2019, Horn et al. discovered that osteoporotic patients had significantly worse radiographic-, clinical-, and complication- or revision-related outcomes compared to those who were not osteoporotic in an 89-patient multicenter retrospective review of patients who received cervical deformity correction surgery (33.3% vs. 9.1%, *p* = 0.021) [19]. Meanwhile, Varshneya et al. investigated factors predicting adverse events in 13,549 patients undergoing surgery for cervical deformity and demonstrated that osteoporotic patients had greater rates of any complications at 90 days compared to non-osteoporotic patients (*p* < 0.001) [12]. The other three investigations did not report an association between osteoporosis and medical complications following surgery (*p* = 0.08, *p* = 0.41, and *p* = 0.809, respectively) [11,18,21]. Smith et al.’s 2016 prospective multicenter study assessed early postoperative complications in the 30 days following cervical deformity surgery in 78 patients, followed by a 2020 study that examined 133 patients with a minimum one-year follow-up [11,18]. In both studies, complications were classified as major or minor depending on the time to resolution and the level of intervention required to rectify the complication. Alternatively, a 2019 study by Passias et al. had a defined list of medical and surgical complications that were used in their study of 123 patients who underwent corrective cervical deformity surgery [21]. These three studies did not observe any significant association between osteoporosis and complication rate.

### 5.2. Reoperation Rate

Varshneya et al. assessed whether there was a relationship between a diagnosis of osteoporosis and the need for revision surgery following cervical spine deformity correction in a 13,549-patient retrospective database study [12]. However, the authors were unable to establish significance and reported an adjusted odds ratio of 0.98 (95%-CI: 0.9–1.1, *p* = 0.720) for patients with osteoporosis requiring a revision procedure within two years [12].

## 6. Discussion

Given the prevalence of osteoporosis as a comorbidity in patients undergoing surgical correction of cervical deformity, this systematic review sought to characterize its impact on postoperative outcomes. Within this context, DJK is a particularly notable outcome of interest, with several studies investigating its potential risk factors. Although two of three investigations into DJK did not establish an association with osteoporosis, the relatively high prevalence of this complication warrants a discussion regarding its risk factors and impact on functional outcomes. Across three studies by Passias et al., DJK was present in 23.1–31.8% of patients and was defined as a Cobb angle > 10° kyphosis from the superior endplate of the end of the fusion construct to the inferior endplate of the second vertebra distal to the construct [9,10,20]. In one of Passias et al.’s investigations, osteoporosis was independently predictive of DJK development [9]. The investigations into risk factors for DJK in cervical ASD surgery have generally been based on small cohorts of less than 150 patients, but a larger matched-cohort study of 1044 patients with thoracolumbar scoliosis (348 with normal BMD, 348 with osteoporosis, and 348 with osteopenia) demonstrated that osteopenic and osteoporotic patients had increased odds of developing DJK (OR 1.88, 95% CI 1.34–2.64), corroborating the idea that reduced bone mineral density predisposes patients to this complication [22].

In general, junctional disorders are thought to be related to under- or overcorrection of the deformity, which leads to abnormal distribution of axial force, causing enhancement of the original kyphosis. Specifically, the longer moment arm at the distal end of the construct compared with the proximal end is subject to greater shear forces as the slope of the transitional segment is increased with greater cervical kyphosis [10,23]. In addition to reporting osteoporosis as a risk factor for DJK, Passias et al. found that DJK was also associated with sagittal malalignment [9]. In a retrospective study of 45 patients who underwent surgical correction of degenerative lumbar scoliosis, Cho et al. identified sagittal decompensation in 8% of patients with proximal junctional issues versus 63% of patients with distal junctional problems, and this discordance is thought to be related to the longer moment arm at the distal end of the instrumented segments [24]. Though there is less overall axial load at the cervical spine in comparison with the lumbar spine, the same principle applies and highlights the importance of careful surgical planning in order to optimize sagittal alignment.

The evidence regarding the relationship between DJK and osteoporosis remains inconclusive, but it is important to understand how this radiographic finding affects long term outcomes. A prospective multicenter study of 67 adult patients who underwent surgical correction of cervical deformity reported a DJK prevalence of 24% within 6 months postoperatively, with 2 of these 16 patients going on to require revision surgery [25]. The authors reported that the second and third most common reasons for distal junctional failure were fracture (19%) and screw pull-out (6%), both of which are at higher risk of development amongst osteoporotic patients [26,27]. However, patient-reported outcomes were not found to be different between DJK and non-DJK groups, a finding that has been substantiated in the current literature, particularly in the long term [28]. In our systematic review, health-related quality of life (HRQOL) scores were similar between DJK and non-DJK cohorts [9,10,20].

Although patient-reported outcome scores do not seem to be affected by DJK, some studies suggest that it may place patients at an increased risk for revision surgery after cervical ASD surgery [23,29]. Alongside an increase in clinical burden, revision surgery in osteoporotic patients has significant financial implications for both patients and hospitals. Numerous studies have explored the impact of osteoporosis on reoperation rates in spine surgery and subsequent increases in medical costs [30,31,32,33]. Although the current literature has yet to describe a cohort specific to cervical deformity within the context of financial burden, these findings suggest that in general, medical management of osteoporosis prior to surgery may be of benefit from a cost-utility perspective. Despite these notions, of the five studies in our review discussing postoperative outcomes and complications, only Varshneya et al. studied revision/reoperation rate independently with regard to osteoporosis and found no direct association (*p* = 0.720) [12]. Though there may still be some indirect relationship between osteoporosis and reoperation rate by means of increased frequency of DJK, the literature remains unclear. However, many studies have suggested that osteoporosis is a significant risk factor for reoperation in cervical degenerative pathology [34,35]. This variation in findings between cervical deformity and degenerative disease may be attributable to the tendency towards extensive stabilization and long-segment fixation present in cervical deformity surgery, as opposed to more localized intervention, often decompression, in degenerative pathology. However, it is important to note that this evidence has yet to be well corroborated and as such, further exploration and evidence of the direct effects of osteoporosis on reoperation rate is needed in cervical deformity surgery prior to concluding on this topic.

In addition to revision and reoperation rates, two of five studies in our review that investigated osteoporosis in the context of overall postoperative complications demonstrated significant associations at 90 days and 1 year following surgery [12,19]. While the existing literature supports the notion that reduced bone mineral density may predispose patients to a variety of complications including junctional failure, pseudarthrosis, and screw loosening, these findings may also be attributable in part to the strict study criteria utilized by Horn et al. [36,37,38]. In this study, an association was observed between osteoporosis and “overall poor outcome,” which was defined as all of the following: developing a complication (including reoperation), poor radiographic outcome, and poor clinical outcome. This was the only study in this review that required all three criteria above to group patients into an “overall poor outcome” category which may have contributed to their statistical findings [19]. Though Varshneya et al. lacked the strict grouping criteria present in the study conducted by Horn et al., the incidence of osteoporosis in their cohort was higher (19.4%) than all of the other studies in this review (10.9–16.7%), which may, in part, contribute to their statistical findings, particularly within the context of their significantly larger sample size (*n* = 13,549) compared to the other studies (all *n* < 150) [12]. Of note, the incidence of osteoporosis of 19.4% in this study more closely resembles that of global prevalence estimates (18–19%), which improves the generalizability of this study as compared to others [7,8]. It remains possible that osteoporosis may influence outcomes and complication rates after corrective surgery for cervical deformity by means of reduced bone mineral density, but the above findings are inconclusive, and more robust analyses with larger samples are necessary prior to any definitive conclusion being made.

### Limitations

While this review explores the notion that osteoporosis may contribute to poorer outcomes in patients undergoing corrective surgery for cervical deformity, it is not without limitations. The studies discussed are largely retrospective in nature, with seven of the eight papers having sample sizes under 150 patients, limiting the strength of their conclusions. Furthermore, the sample size of osteoporotic patients within these studies largely fall under 20. There were no randomized controlled trials that met the search criteria for this systematic review; however, the level of evidence of each of the included studies are outlined within the tables above. This presents a significant limitation to the strength and generalizability of the findings. In addition, the global prevalence of osteoporosis has been approximated at 18–19%, whereas the majority of studies included in this review largely fell below that estimate, limiting the generalizability of the results [7,8]. Given the significant overlap of authors and institutions in the studies reviewed, a meta-analysis was unable to be reliably performed as duplicate records between studies could not be parsed out. Additionally, the presence of only eight studies fulfilling the search criteria was, in itself, a major limitation of this review. Variations in timing of postoperative complications (i.e., 30 days, 90 days, 1 year, etc.) further contributed to the challenge in conducting an accurate meta-analysis. Future studies should explore larger samples, which may yield more representative frequencies of osteoporosis and improve statistical power. In addition, variables such as sarcopenia and its interplay with osteoporosis and complication rates should be given consideration in future studies. Another limitation of this study stems from its focus on cervical deformity as a pathology rather than a procedural (i.e., ACDF)-focused analysis. While studies encompassing large cohorts of patients undergoing individual cervical procedures such as an ACDF exist, those include patients with both deformity and degenerative pathology and were not included in this study. While the objective of this study was to focus on cervical deformity, future studies should consider exploring the effects of osteoporosis in the cervical spine on in a procedure-based fashion. Overall, given the prevalence of osteoporosis as a comorbidity in patients undergoing surgical correction of cervical deformity, alongside the lack of standardized guidelines for preoperative screening and management of osteoporosis in these patients, this review highlights the importance of further research exploring the interplay between bone mineral density and postoperative outcomes.

## 7. Conclusions

The findings of this systematic review do not suggest that osteoporosis is predictive of the need for reoperation or the development of postoperative complications including DJK, among others. However, better powered studies with more representative proportions of osteoporotic patients that reflect global prevalence are needed to assess less-frequent complications and outcomes. Though there is evidence suggesting that reduced bone mineral density may result in poorer postoperative outcomes, the majority of studies did not determine an association in patients undergoing corrective surgery for cervical deformity. These findings highlight the need for further investigation into the impact of osteoporosis on surgical outcomes prior to making definitive recommendations for perioperative screening and management.

## Figures and Tables

**Figure 1 jcm-14-06196-f001:**
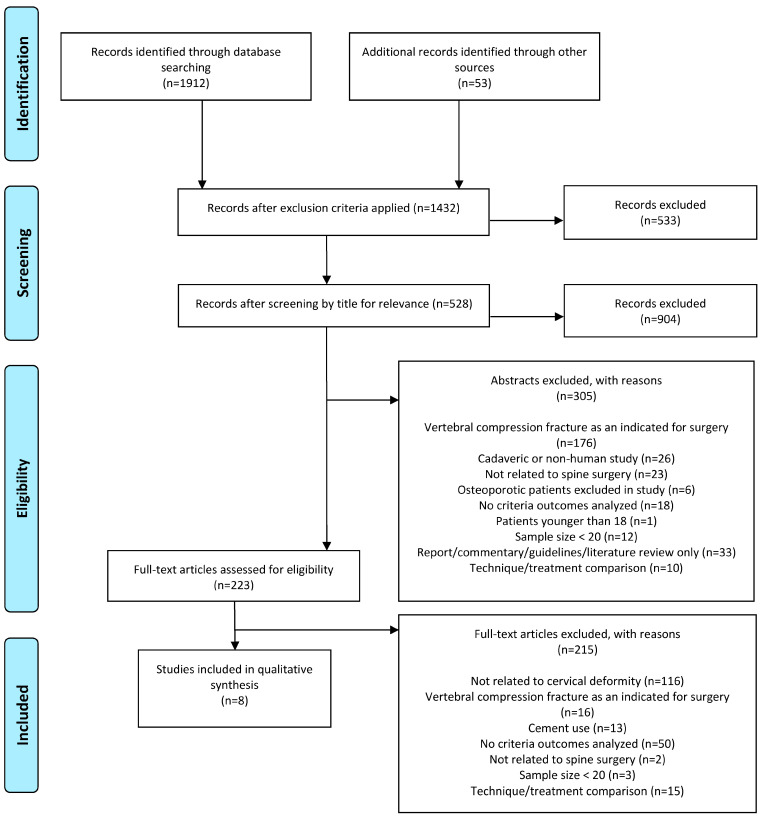
PRISMA flowsheet for the process of study selection.

**Figure 2 jcm-14-06196-f002:**
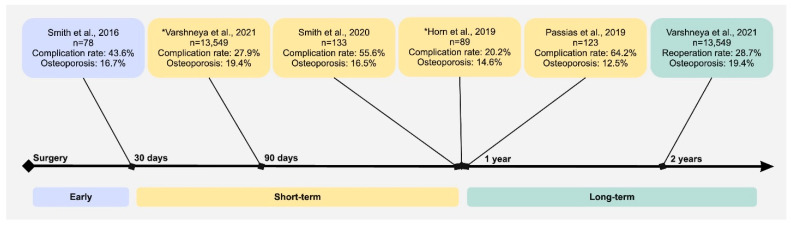
Timeline displaying frequency of osteoporosis and outcome of interest in each study included in the review at various timepoints [11,12,18,19,20]. * indicates a study which demonstrated a statistically significant association between osteoporosis and outcome of interest.

**Table 1 jcm-14-06196-t001:** Characteristics of studies focusing on surgery for cervical deformity.

Study	Type of Study	Patient Population/ Indications	Radiographic Outcomes	Complications	Level of Evidence	Quality of Evidence (Grade)
Smith et al., 2016 [11]	Retrospective review of prospectively collected database (*n* = 78)	Patients undergoing fusion for cervical deformity.		Any early post-operative complications within 30 days.	III	Very low
Smith et al., 2020 [18]	Retrospective review of prospectively collected database (*n* = 133)	Patients undergoing fusion for cervical deformity.		Any major or minor post-operative complications within 1 year.	III	Very low
Horn et al., 2019 [19]	Retrospective review of prospectively collected database (*n* = 89)	Patients undergoing fusion for cervical deformity.		Overall poor outcome (defined as meeting all three of the following criteria: needing a reoperation/developing a complication, poor radiographic outcome, and poor clinical outcome) within 1 year.	III	Very low
Passias et al., 2020 [20]	Retrospective review of prospectively collected database (*n* = 117)	Patients undergoing fusion for cervical deformity.	Distal junctional kyphosis.		III	Very low
Passias et al., 2023 [9]	Retrospective review of prospectively collected database (*n* = 110)	Patients undergoing fusion for cervical deformity.	Distal junctional kyphosis.		III	Very low
Passias et al., 2018 [10]	Retrospective review of prospectively collected database (*n* = 101)	Patients undergoing long fusion for cervical deformity.	Distal junctional kyphosis.		III	Very low
Passias et al., 2019 [21]	Retrospective review of prospectively collected database (*n* = 123)	Patients undergoing long fusion for cervical deformity.		Overall post-operative complications within 1 year.	III	Very low
Varshneya et al., 2021 [12]	Retrospective database review (*n* = 13,549)	Patients undergoing multilevel primary fusion for adult cervical deformity.		Any post-operative complications within 90 days. Any reoperation within 2 years.	III	Low

**Table 2 jcm-14-06196-t002:** Key details and findings of studies focusing on surgery for cervical deformity.

Study	Type of Study	Patient Population/ Indications	Outcome of Interest	Frequency of Outcome of Interest	Frequency of Osteoporosis in Cohort	Key Findings
Smith et al., 2016 [11]	Retrospective review of prospective multicenter database (*n* = 78)	Patients undergoing fusion for cervical deformity.	Any early post-operative complications within 30 days.	34 (43.6%)	13 (16.7%)	No association between osteoporosis and early complications both major and minor (*p* = 0.08).
Smith et al., 2020 [18]	Retrospective review of prospective multicenter database (*n* = 133)	Patients undergoing fusion for cervical deformity.	Any major or minor post-operative complications within 1 year.	74 (55.6%)	22 (16.5%)	No association between osteoporosis and complications both major and minor within 1 year postoperatively (*p* = 0.41).
Horn et al., 2019 [19]	Retrospective review of prospective multicenter database (*n* = 89)	Patients undergoing fusion for cervical deformity.	Overall poor outcome (defined as meeting all three of the following criteria: needing a reoperation/developing a complication, poor radiographic outcome, and poor clinical outcome) within 1 year.	18 (20.2%)	13 (14.6%)	Osteoporotic patients (33.3%) had worse radiographic, clinical, and complication or revision related outcomes compared to non-osteoporotic patients (9.9%) (*p* = 0.021) at 1 year postoperatively.
Passias et al., 2020 [20]	Retrospective review of prospective multicenter database (*n* = 117)	Patients undergoing fusion for cervical deformity.	Distal junctional kyphosis.	27 (23.1%)	15 (13.2%)	No association between osteoporosis and distal junctional kyphosis.
Passias et al., 2023 [9]	Retrospective review of prospective multicenter database (*n* = 110)	Patients undergoing fusion for cervical deformity.	Distal junctional kyphosis.	35 (31.8%)	NA	Osteoporosis increased the likelihood of distal junctional kyphosis postoperatively with a coefficient of 0.633 in a predictive model with an AUC of 77.8%.
Passias et al., 2018 [10]	Retrospective review of prospective multicenter database (*n* = 101)	Patients undergoing long fusion for cervical deformity.	Distal junctional kyphosis.	24 (23.8%)	11 (10.9%)	No association between osteoporosis and distal junctional kyphosis.
Passias et al., 2019 [21]	Retrospective review of prospective multicenter database (*n* = 123)	Patients undergoing long fusion for cervical deformity.	Overall post-operative complications within 1 year.	79 (64.2%)	15 (12.5%)	No association between osteoporosis and overall complications within 1 year postoperatively (*p* = 0.809).
Varshneya et al., 2021 [12]	Retrospective database (*n* = 13,549)	Patients undergoing multilevel primary fusion for adult cervical deformity.	Any post-operative complications within 90 days. Any reoperation within 2 years.	3785 (27.9%) 3893 (28.7%)	2630 (19.4%)	Osteoporotic patients had a greater rate of any complication at 90 days postoperatively (*p* < 0.001). No association between osteoporosis and revision surgery (*p* = 0.720).

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
