# Peer review of "A Systematic Review of the Effect of Osteoporosis on Radiographic Outcomes, Complications, and Reoperation Rate in Cervical Deformity"

_jcm, 2025, doi:10.3390/jcm14176196_

Round 1

Reviewer 1 Report

Comments and Suggestions for Authors

Review -- A Systematic Review of the Effect of Osteoporosis on 2 Radiographic Outcomes, Complications, and Reoperation Rate 3 in Cervical Deformity

Thank you for the opportunity to review this paper which was read with great interest. The authors have performed a review of the current literature available concerning the effects of osteoporosis on patients with cervical deformity. The subject matter is highly relevant. The use of a systematic review design is appropriate, and this article will be a welcome addition to the cervical deformity literature. 

Introduction: 
This is written well and sets the foundation for the review’s subject matter.

Methods: 
The databases searched and keywords used are appropriate. The exclusion criteria are reasonable and don’t appear to bias the search strategy. The methodology behind quality assessment, data extraction and statistical analysis is appropriate.

Results:
The results are well presented. It was interesting to observe the paucity of studies significantly associating osteoporosis with DJK. This article suggests that more of such investigations are much needed. The graphic illustrating the osteoporosis frequency in different studies is well-informed and is a highly beneficial inclusion.

Discussion:
The discussion highlights the notable lack of evidence for the association between osteoporosis and outcomes in cervical deformity patients. The authors have made efforts to summarise the currently available evidence and highlight the need for studies of higher quality and power in order to more confidently infer conclusions. The article’s conclusion further reinforces this.

Overall:
The authors should be congratulated for this research effort. There is notable paucity in the available literature in this context, and such research endeavours are needed. This article would be a welcome and valuable addition to the current evidence concerning cervical deformity.

Author Response

Comment 1: "Thank you for the opportunity to review this paper which was read with great interest. The authors have performed a review of the current literature available concerning the effects of osteoporosis on patients with cervical deformity. The subject matter is highly relevant. The use of a systematic review design is appropriate, and this article will be a welcome addition to the cervical deformity literature."

Response 1: Thank you for your comment.

Change to text: N/A

Comment 2: "Introduction: 
This is written well and sets the foundation for the review’s subject matter."

Response 2: Thank you for your comment.

Change to text: N/A

Comment 3: "Methods: 
The databases searched and keywords used are appropriate. The exclusion criteria are reasonable and don’t appear to bias the search strategy. The methodology behind quality assessment, data extraction and statistical analysis is appropriate."

Response: Thank you for your comment.

Change to text: N/A

Comment 4: "Results:
The results are well presented. It was interesting to observe the paucity of studies significantly associating osteoporosis with DJK. This article suggests that more of such investigations are much needed. The graphic illustrating the osteoporosis frequency in different studies is well-informed and is a highly beneficial inclusion."

Response: Thank you for your comment.

Change to text: N/A

Comment 5: "Discussion:
The discussion highlights the notable lack of evidence for the association between osteoporosis and outcomes in cervical deformity patients. The authors have made efforts to summarise the currently available evidence and highlight the need for studies of higher quality and power in order to more confidently infer conclusions. The article’s conclusion further reinforces this."

Response: Thank you for your comment.

Change to text: N/A

Comment 6: "Overall:
The authors should be congratulated for this research effort. There is notable paucity in the available literature in this context, and such research endeavours are needed. This article would be a welcome and valuable addition to the current evidence concerning cervical deformity."

Response: Thank you for your comment.

Change to text: N/A

Reviewer 2 Report

Comments and Suggestions for Authors

The authors present a well-structured review article on the possible relationship between osteoporosis as a risk factor and poor outcomes among patients undergoing surgery for cervical spine deformity.  The methods for article selection are clearly described, and the relevant information from the selected studies is presented and interpreted in a clear and appropriate way.

The concerns about the paper are in a two or three domains. First, the authors do not present compelling data on the prevalence of cervical deformity as an indication or underlying diagnosis for surgery.  Since many articles about spine surgery use more specific terms like kyphosis, it's not clear how large a subset of the overall domain of spine surgery the authors are studying.   What proportion of a typical busy spine surgeon's practice might fall under the scope of this review?

The small sample sizes of the eight studies selected for review suggests that this is perhaps not a very common problem or surgical situation, and the relatively low reported prevalence of osteoporosis in the studies cited make the conclusions that are drawn come from a sample of perhaps 20 or fewer patients in a given study who have osteoporosis as a risk factor.

The results are mixed, and according to the authors' classification of study quality, the studies are all relatively low-quality studies.  In this domain, it is clearly not possible to have RCT-level evidence, but the combination of mixed findings or non-significant findings in specific studies and low-quality evidence suggests that we just can't learn much from this review.  It's not the authors' fault - they did the best they could with the evidence available, but it appears that the evidence just isn't that clear or strong.

The authors also don't make a compelling case for why evidence from studies of lumbar spine surgery or studies of cervical surgery based on procedure type rather than diagnosis (e.g., Pinter et al, World Neurosurgery 2022) wouldn't be relevant.  The Pinter study focuses of ACDF as a procedure rather than deformity as an underlying pathology, but wouldn't results of a study like theirs be about as informative as those of the eight studies selected?  Why would one expect a different effect of osteoporosis in cervical deformity surgery than one might see in either other classes of cervical surgery or surgery in other parts of the spine?

The authors acknowledge many of these problems in their limitation section, but it seems like it might be possible to make a more compelling cases for a focus specifically on deformity surgery rather than cervical spine surgery more generally, or perhaps have a richer data set to work with if the net were cast a little more broadly.

Author Response

Comment 1: "The authors present a well-structured review article on the possible relationship between osteoporosis as a risk factor and poor outcomes among patients undergoing surgery for cervical spine deformity.  The methods for article selection are clearly described, and the relevant information from the selected studies is presented and interpreted in a clear and appropriate way."

Response 1: Thank you for your comment.

Change to text: N/A

Comment 2: "The concerns about the paper are in a two or three domains. First, the authors do not present compelling data on the prevalence of cervical deformity as an indication or underlying diagnosis for surgery.  Since many articles about spine surgery use more specific terms like kyphosis, it's not clear how large a subset of the overall domain of spine surgery the authors are studying.   What proportion of a typical busy spine surgeon's practice might fall under the scope of this review?"

Response 1: Thank you for your comment. We agree that further contextualization of the term cervical deformity would benefit this manuscript. While it is difficult to estimate the proportion of a spine surgeon's practice that might encompass cervical deformity due to surgeon and institutional variation in case selection, patient population, and practice characteristics, we have added further info regarding incidence to the manuscript. 

Change to text: Lines 42-43: "Cervical deformity encompasses a variety of spinal pathology involving the abnormal curvature or misalignment of the cervical spine with an estimated incidence of approximately 30%"

Comment 3: "The small sample sizes of the eight studies selected for review suggests that this is perhaps not a very common problem or surgical situation, and the relatively low reported prevalence of osteoporosis in the studies cited make the conclusions that are drawn come from a sample of perhaps 20 or fewer patients in a given study who have osteoporosis as a risk factor."

Response 1: Thank you for your comment. We agree that the sample sizes in many of the studies included are relatively low with the exception of Varshneya et al. (n=2,630). As such, we have reiterated sample size as a limitation within the manuscript as below:

Change to text: Lines 250-251: "Furthermore, the sample size of osteoporotic patients within these studies largely fall under 20."

Comment 4: "The results are mixed, and according to the authors' classification of study quality, the studies are all relatively low-quality studies.  In this domain, it is clearly not possible to have RCT-level evidence, but the combination of mixed findings or non-significant findings in specific studies and low-quality evidence suggests that we just can't learn much from this review.  It's not the authors' fault - they did the best they could with the evidence available, but it appears that the evidence just isn't that clear or strong."

Response 1: Thank you for your comment. We agree that the evidence is limited surrounding this subject and we have attempted to address this in the limitations section. 

Change to text: Lines 250-251: "Furthermore, the sample size of osteoporotic patients within these studies largely fall under 20."

Comment 5: "The authors also don't make a compelling case for why evidence from studies of lumbar spine surgery or studies of cervical surgery based on procedure type rather than diagnosis (e.g., Pinter et al, World Neurosurgery 2022) wouldn't be relevant.  The Pinter study focuses of ACDF as a procedure rather than deformity as an underlying pathology, but wouldn't results of a study like theirs be about as informative as those of the eight studies selected?  Why would one expect a different effect of osteoporosis in cervical deformity surgery than one might see in either other classes of cervical surgery or surgery in other parts of the spine?"

Response 1: Thank you for your comment. We agree that studies such as Pinter et al. provide valuable information focusing on ACDF as a procedure. Pinter et al. included all cases of ACDF involving degenerative pathologies at a single institution and as such may include cases outside of deformity. The primary objective of this study was to focus on cervical deformity as a pathology however we do agree that procedure-wise study of this subject may provide valuable insights into surgical management of osteoporotic patients. We have added further discussion on this topic as follows:

Change to text: Lines 261-267: "Another limitation of this study stems from its focus on cervical deformity as a pathology rather than a procedural (i.e. ACDF) focused analysis. While studies encompassing large cohorts of patients undergoing individual cervical procedures such as an ACDF exist, those include patients with both deformity and degenerative pathology and were not included in this study. While the objective of this study was to focus on cervical deformity, future studies should consider exploring the effects of osteoporosis in the cervical spine on in a procedure-based fashion."

Comment 6: "The authors acknowledge many of these problems in their limitation section, but it seems like it might be possible to make a more compelling cases for a focus specifically on deformity surgery rather than cervical spine surgery more generally, or perhaps have a richer data set to work with if the net were cast a little more broadly."

Response 1: Thank you for your comment. We agree that the manuscript could use further justification for its focus on cervical deformity rather than encompassing all of cervical spine surgery. We have added further discussion to the manuscript as follows:

Change to text: Lines 261-267: "Another limitation of this study stems from its focus on cervical deformity as a pathology rather than a procedural (i.e. ACDF) focused analysis. While studies encompassing large cohorts of patients undergoing individual cervical procedures such as an ACDF exist, those include patients with both deformity and degenerative pathology and were not included in this study. While the objective of this study was to focus on cervical deformity, future studies should consider exploring the effects of osteoporosis in the cervical spine on in a procedure-based fashion."

Reviewer 3 Report

Comments and Suggestions for Authors

This nice work examines the role of osteoporosis in complications rates after cervical spine surgery. 

Cervical spine receives less forces and weight than the lumbar, so the effect of osteoporosis on these bones is poorly investigated until now. 

The authors manage to provide in a few words a nice summary of the existing literature without tiring the reader. The conclusions drawn are well argumented and the included papers are well analyzed in the text. 

Maybe an extra sentence to the introduction or discussion about the increased complication rate in lumbar spine surgeries in those patients would be nice. The comparison will increase the impact of the conclusions drawn.

What is more, it would be interesting, although probably not possible, to add another variable to the statistic - sarcopenia. It is more and more reported that sarcopenia increases the risk for complications and in combination to osteoporosis could be disastrous for patients and surgeons. 

Otherwise I find it very nice, well done.

Author Response

Comment 1: "This nice work examines the role of osteoporosis in complications rates after cervical spine surgery."

Response 1: Thank you for your comment.

Change to text: N/A

Comment 2: "Cervical spine receives less forces and weight than the lumbar, so the effect of osteoporosis on these bones is poorly investigated until now."

Response 1: Thank you for your comment. We agree that this is an important point and have added it to the discussion.

Change to text: Line 58-60: "However, given the reduced load present in the cervical spine as compared to the lumbar spine and the resultant varying complication rates, the effect of osteoporosis may vary as well. "

Comment 3: "The authors manage to provide in a few words a nice summary of the existing literature without tiring the reader. The conclusions drawn are well argumented and the included papers are well analyzed in the text."

Response 1: Thank you for your comment.

Change to text: N/A

Comment 4: "Maybe an extra sentence to the introduction or discussion about the increased complication rate in lumbar spine surgeries in those patients would be nice. The comparison will increase the impact of the conclusions drawn."

Response 1: Thank you for your comment. We agree additional contextualization of these findings relative complication rates in lumbar spine surgeries would benefit the manuscript. We have added this to the manuscript.

Change to text: Lines 58-60: "However, given the reduced load present in the cervical spine as compared to the lumbar spine and the resultant varying complication rates, the effect of osteoporosis may vary as well."

Comment 5: "What is more, it would be interesting, although probably not possible, to add another variable to the statistic - sarcopenia. It is more and more reported that sarcopenia increases the risk for complications and in combination to osteoporosis could be disastrous for patients and surgeons."

Response 1: Thank you for your comment. We agree that sarcopenia would be a valuable addition to the study. Unfortunately given the stringent criteria for the literature search it is not feasible to include without altering the methodology of the manuscript. However, we have added further discussion on the importance and relevance of sarcopenia for future studies on this subject.

Change to text: Lines 259-260: "In addition, variables such as sarcopenia and its interplay with osteoporosis and complication rates should be given consideration in future studies. "

Round 2

Reviewer 2 Report

Comments and Suggestions for Authors

Thea authors have addressed the comments or suggestions made in the initial round of review, but the changes are minimal and do not create any significant difference in the manuscript.  A sentence or two has been added to the Intro or to the Discussion, mainly to more clearly acknowledge limitations of the approach (e.g., focus on deformity) or the available literature (small sample sizes).

As a result, the revised paper is not more informative to readers than the original paper was.  Surgeons or other readers will not be left with a clearer sense of the possible impact of osteoporosis on cervical spine surgery outcomes than they had before.  They will learn that a small set of low-quality, small-sample studies have produced equivocal results.